# Bring Future Vision: Dynamic Computation Allocation Guided by Lightweight Feature Forecaster

## Abstract

The deployment of large language models (LLMs) in practical scenarios is hampered by their low inference efficiency. While token-wise computation allocation emerges as a promising solution, existing methods suffer from irreversible information loss and suboptimal token selection due to the greedy routing paradigm. This paper introduces a novel paradigm, informed routing, which proactively addresses these limitations. Our key insight is to employ Lightweight Feature Forecaster (LFF) — simple, low-cost networks that learn to approximate the transformations of individual model components — before making any routing decisions. This allows the router to assess a token's recoverability (ease of approximation) rather than just its immediate importance. Extensive experiments demonstrate that our approach achieves state-of-the-art performance across various sparsity levels on language modeling and reasoning tasks. Notably, even without final LoRA fine-tuning, our method matches or surpasses strong baselines that require full fine-tuning, all while reducing training time by over 50%.

## 1 Introduction

The emergence of large language models (LLMs) has catalyzed breakthroughs across diverse industries, from code generation (OpenAI et al., 2024; Rozière et al., 2024) to scientific discovery (Zheng et al., 2025). Scaling laws have established computational requirements as a primary bottleneck in the development and deployment of LLMs (Kaplan et al., 2020). Therefore, reducing this computational overhead has become a key research objective.

Early work primarily focused on **static pruning** methods, which permanently remove a fixed subset of parameters or components from the model (Han et al., 2016; Ma et al., 2023). While effective for compression, these approaches fail to exploit the varying importance of tokens during inference. More recently, the observation of diverse token criticality has motivated a shift toward **dynamic computation allocation (DCA)** (Raposo et al., 2024), where different tokens undergo different amounts of computation. DCA partitions the model into computational units—ranging from coarse-grained layers to finer-grained sub-layer components (e.g., self-attention blocks and feed-foward network blocks within a single layer), each equipped with a router. These routers, typically small MLPs, are trained post-hoc to decide whether to execute or skip a unit for each token. In practice, important tokens are routed through most of the model's parameters, while less important ones can skip substantial computation. This flexibility mirrors human language processing, where critical words are analyzed in depth while less informative ones receive only shallow processing.

However, existing DCA methods are constrained by a paradigm we term **greedy routing**. Routers are trained to make a simple, binary choice: fully execute a computational unit or skip it entirely. Performance recovery is then attempted via lightweight fine-tuning (e.g., LoRA (Hu et al., 2022)). [1] The decision to skip is based on minimizing the *immediate* performance drop, without considering the long-term consequences (Zhao et al., 2025). This greedy approach suffers from two fundamental flaws:

---

[1] Some works have also explored jointly training the router and the recovery module, however, empirical evidence suggests that this can lead to performance degradation (Zhao et al., 2025).

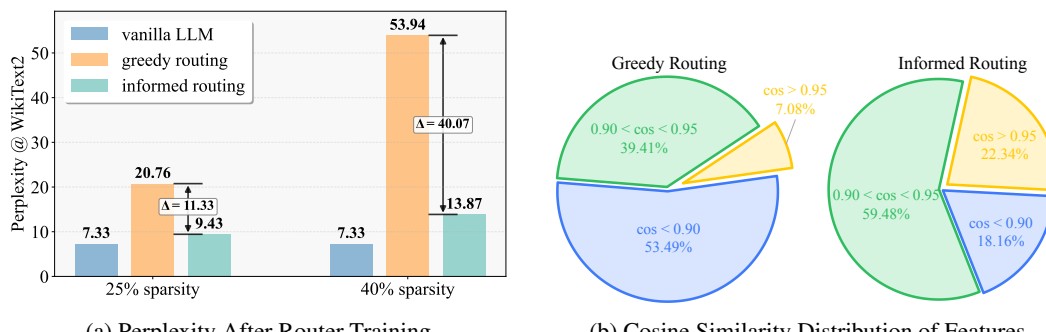

(a) Perplexity After Router Training  (b) Cosine Similarity Distribution of Features

Figure 1: The limitation of greedy routing and the promise of informed routing. Under the same sparity ratio, informed routing (a) reduces the perplexity and (b) increases feature similarity.

1. **The All-or-Nothing Dilemma:** By forcing a rigid execute-or-skip decision, this paradigm offers no middle ground. Skipping a unit causes irreversible information loss, disrupting the model's internal feature distributions and requiring costly fine-tuning to recover performance. As shown in Figure 1a, skipping tokens leads to a significant increase in perplexity, from 7.33 to 20.76 (25% sparsity) and 53.94 (40% sparsity).

2. **Short-Sighted Token Selection:** The router's focus on immediate impact is a poor proxy for true importance. A token that causes a large immediate drop when skipped is not necessarily indispensable; its transformation might be simple and easily recoverable later. Conversely, a token with low immediate impact might be crucial for maintaining subtle, long-range dependencies that are difficult to restore once lost.

To overcome these limitations, we propose a paradigm shift from greedy decision-making to **informed routing**. Our central idea is to replace the binary execute-or-skip choice with a more nuanced execute-or-approximate decision. We achieve this by equipping each computational unit with a **Lightweight Feature Forecaster (LFF)**—a small, efficient network trained to mimic the output of its larger counterpart. The router's task is no longer to guess which tokens can be safely dropped, but to determine which tokens are *predictable*. If a token's transformation can be accurately approximated by the LFF, it is routed through this efficient path. If the transformation is too complex to be forecasted, the token is processed by the original, powerful unit.

This "informed" approach allows the router to learn a policy based on a token's **recoverability**, not just its immediate impact. For instance, an analysis of feature similarity (Figure 1b) reveals that LFF increases the proportion of features that remain highly similar (cosine similarity $> 0.95$), from 7.08% to 22.34%. Such significant increase provides concrete evidence for the existence of a substantial fraction of *predictable* tokens—tokens whose transformations can be accurately approximated by the LFF. Based on this insight, we propose a simple three-stage pipeline: (1) train LFF to approximate their corresponding units, (2) train routers to choose between the original unit and its LFF for each token, and (3) perform optional, lightweight LoRA (Hu et al., 2022) fine-tuning to polish the final model. Crucially, throughout this process, the router and LFF parameters are deliberately configured to ensure a fair comparison with the baseline experiments.

Our contributions are as follows:

1. We identify the core limitations of the prevailing **greedy routing** paradigm in DCA, namely its rigid all-or-nothing mechanism and its short-sighted reliance on immediate impact as a routing criterion.

2. We introduce **informed routing**, a new paradigm enabled by **Lightweight Feature Forecaster (LFF)**, which replaces skipping with efficient approximation and allows routing decisions to be based on a token's recoverability.

3. We demonstrate through extensive experiments that our method achieves state-of-the-art efficiency, significantly reducing training overhead and improving final performance. Our analysis further reveals that self-attention modules are highly "predictable", making them prime candidates for approximation.

## 2 RELATED WORKS

**Static Pruning** Broadly speaking, static pruning techniques related to our approach fall into two main categories: token pruning and parameter pruning. **Static Token Pruning** methods identify and remove tokens deemed redundant, and pruned tokens bypass subsequent transformer layers, which are widely used in Vision-Language Models(VLMs). SpecVLM(Ji et al., 2025) introduces a 'verifier' model to estimate the importance of video tokens. VisionDrop(Xu et al., 2025) identifies token importance via intra-modal attention. In contrast, our dynamic approach achieves flexibility along *model depth*, allowing each token to undergo full computation only in the layers where it is most needed, thereby preserving information while adaptively saving computation. **Static Parameter Pruning** techniques permanently remove fixed structural components(e.g., layers or neurons), resulting in a uniformly smaller model. SliceGPT(Ashkboos et al., 2024) employs Principal Component Analysis (PCA) on the orthogonally transformed parameters, followed by the removal of entire rows/columns. Shortened-llama(Kim et al., 2024) demonstrates that depth pruning is more efficient than width for LLM inference. ShortGPT(Men et al., 2024) proposes Block Influence(BI) to quantitatively estimate the importance of layers in large language models, and subsequently prunes the less important layers. LLM-Streamline(Chen et al., 2025a) removes consecutive layers and then replaces them with a smaller model. Parameter reduction in capacity applies inflexibly to all tokens, regardless of their importance. Conversely, the proposed dynamic method provides flexibility across *input token sequence*, i.e. for a given parameter structure, only a subset of tokens utilizes it while others bypass it via a lightweight path (LFF), enabling a more granular and input-adaptive efficiency.

**Dynamic Computation Allocation** Dynamic computation allocation methods leverage the observation that linguistic representations evolve at varying paces across tokens. Central to these techniques is the *router* mechanism—a lightweight classifier that dynamically assigns computational paths to tokens. The router acts as a per-token binary classifier at each computation unit. For every token representation, it first computes skip/keep probabilities using a multilayer perceptron. The execution path is then determined via argmax sampling: if "skip" is selected, the token bypasses the unit and remains unchanged; if "keep" is chosen, it undergoes transformation within the unit. This gating mechanism results in dynamic, token-wise computation graphs where inactive tokens propagate directly to the next layer without being processed. Contemporary approaches to dynamic computation allocation exhibit notable variations in sparsity control and computational granularity. Mixture-of-Depths (MoD) (Raposo et al., 2024), enforces a fixed sparsity ratio per layer block, which limits its ability to adapt to input-specific redundancy patterns and ultimately constrains its efficiency. In contrast, subsequent work such as D-LLM (Jiang et al., 2024) introduces global adaptive sparsity, dynamically allocating computation across layers in response to input characteristics. Building on this, SkipGPT (Zhao et al., 2025) further refines the granularity by decoupling attention and MLP operations within each layer. It employs separate routers to independently skip each submodule, enabling more flexible computation paths while maintaining globally optimized sparsity. However, all of these are built with greedy routing.

**Error Compensation** Prior works have explored error compensation to mitigate compression-induced accuracy drops. RECAP (Lee et al., 2025) transfers the statistics of pruned channels to adjacent weights. PRUNE&COMP (Chen et al., 2025b) rescales remaining weights offline to compensate for the magnitude gap after layer removal. Olica (He & Lin, 2025) introduces a linear mapping for low-rank compensation in FFNs. While these methods perform *weight-wise* or *channel-wise* compensation, our approach operates in a *token-wise* manner. We dynamically route tokens to either the original model or a LFF, enabling finer-grained and adaptive error recovery. This allows critical tokens to retain full precision while approximating redundant ones, achieving more flexible accuracy-efficiency trade-offs.

## 3 METHODOLOGY

In this section, we introduce **informed routing**, a framework that replaces rigid skip decisions with efficient approximations. Figure 2 provides an overview of the framework, and we now detail the architecture, the LFF design, and the training procedure.

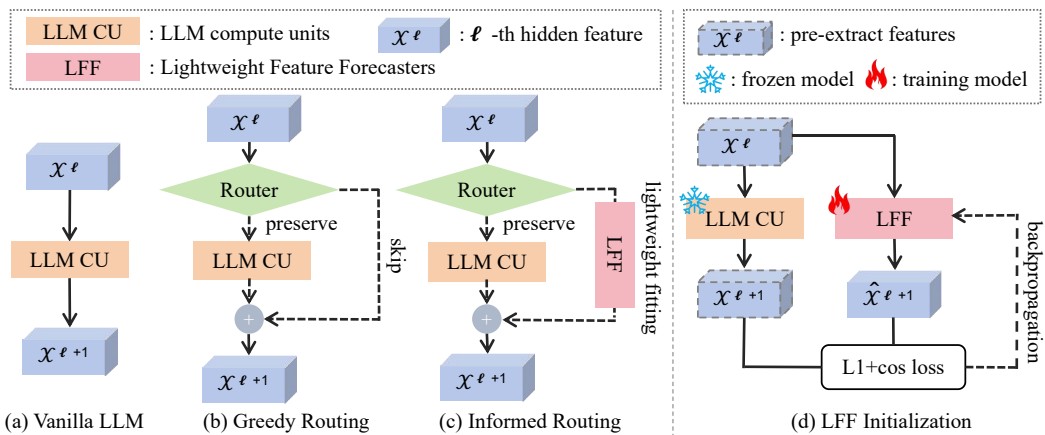

Figure 2: (a), (b), and (c) present the architectural comparison diagrams of the vanilla LLM, greedy routing, and our proposed informed routing paradigm. (d) illustrates how the LFF initialization is performed.

## 3.1 PRELIMINARIES AND NOTATION

Consider a transformer-based LLM with $L$ layers. For layer $\ell \in \{1, \ldots, L\}$, let $\mathbf{X}^\ell \in \mathbb{R}^{N \times d}$ denote the input token embeddings, where $N$ is sequence length and $d$ is the hidden dimension. Each transformer layer $\ell$ processes the input through two components, i.e. self-attention and feed-forward network(FFN), with pre-normalization and residual connections:

$$\mathbf{X}_{\text{att}}^\ell = \mathcal{A}^\ell \left( \text{Norm}(\mathbf{X}^\ell) \right) + \mathbf{X}^\ell \quad \text{(Self-attention module)} \tag{1}$$

$$\mathbf{X}^{\ell+1} = \mathcal{F}^\ell \left( \text{Norm}(\mathbf{X}_{\text{att}}^\ell) \right) + \mathbf{X}_{\text{att}}^\ell \quad \text{(FFN module)} \tag{2}$$

where: $\mathcal{A}^\ell$: Multi-head self-attention at layer $\ell$, $\mathcal{F}^\ell$: Feed-forward network at layer $\ell$, $\mathbf{X}_{\text{att}}^\ell$: Intermediate representation after attention, $\mathbf{X}^{\ell+1}$: Output embeddings serving as input to layer $\ell + 1$.

Following SkipGPT's granularity, we decompose each transformer layer $\ell$ into two *computational units*: $\mathcal{U}^{\ell,\text{SA}}$ (self-attention) and $\mathcal{U}^{\ell,\text{FFN}}$ (feed-forward network). For each unit, a lightweight *router* $\mathcal{R}^{\ell,k} : \mathbb{R}^d \rightarrow \mathbb{R}^2$ (where $k \in \{\text{SA}, \text{FFN}\}$) makes token-wise pruning decisions. The router is implemented as a two-layer MLP with bottleneck dimension, i.e $\mathbb{R}^d \rightarrow \mathbb{R}^{\lfloor d/4 \rfloor} \rightarrow \mathbb{R}^2$.

The router outputs decision logits for each token $\mathbf{x}_i^{\ell,k}$:

$$\mathbf{r}_i^{\ell,k} = \mathcal{R}^{\ell,k}(\mathbf{x}_i^{\ell,k}) \in \mathbb{R}^2 \tag{3}$$

with routing probabilities obtained via softmax:

$$p_i^{\ell,k} = \sigma(\mathbf{r}_i^{\ell,k})_1 = \frac{\exp(\mathbf{r}_i^{\ell,k}[1])}{\sum_{c=0}^1 \exp(\mathbf{r}_i^{\ell,k}[c])} \tag{4}$$

where class $c = 1$ indicates *preserving precision through original LLM compute unit* and $c = 0$ indicates *lightweight fitting via LFF branch*. Figure 2(c) illustrates this computational flow.

During the forward pass, a hard binary mask $\mathbf{p}^{\ell,k}$ is sampled by applying the $\arg\max$ operation to Gumbel-Softmax logits, producing discrete $0, 1$ values. In the backward pass, the gradient is estimated using a continuous softmax approximation with temperature $\tau$, enabling differentiable training. The temperature is annealed linearly from $\tau = 5.0$ to $\tau = 1.0$ to sharpen the distribution over time. Modern frameworks such as PyTorch provide built-in functions like $F.gumbel\_softmax$, which facilitates end-to-end training of discrete latent variable models.

The training objective minimizes computation while preserving performance by enforcing a target relative sparsity $S_{\text{target}}$ (e.g., 50%). The global computation fraction is:

$$\rho = \frac{1}{2LN} \sum_{\ell=1}^L \sum_{k \in \{\text{SA},\text{FFN}\}} \|\mathbf{p}^{\ell,k}\|_0 \tag{5}$$

where $\|\mathbf{p}^{\ell,k}\|_0 = \sum_i p_i^{\ell,k}$, and we regulate $\rho$ toward $S_{\text{target}}$ during training (Section 3.3).

## 3.2 LIGHTWEIGHT FEATURE FORECASTER

The core innovation of our paper is the **lightweight feature forecaster** $\mathcal{F}^{\ell,k} : \mathbb{R}^d \to \mathbb{R}^d$ that approximates the input-output mapping of computational unit $\mathcal{U}^{\ell,k}$ *before* routing decisions. This architectural shift transitions the routing paradigm from reactive recovery to proactive preservation. For efficiency, $\mathcal{F}^{\ell,k}$ uses a bottleneck architecture:

$$\mathcal{F}^{\ell,k}(\mathbf{x}) = \mathbf{W}_2^{\ell,k} \cdot \left( \mathbf{W}_1^{\ell,k}\mathbf{x} + \mathbf{b}_1^{\ell,k} \right) + \mathbf{b}_2^{\ell,k} \tag{6}$$

where $\mathbf{W}_1^{\ell,k} \in \mathbb{R}^{r \times d}$, $\mathbf{W}_2^{\ell,k} \in \mathbb{R}^{d \times r}$ with $r \ll d$ (e.g., $r = 100$ with $d = 4096$ for Llama3.1-8B (Grattafiori et al., 2024)). This yields minimal parameters: $4096 \times 100 + 100 \times 4096 \approx 0.82$M ($0.02\%$ of $\mathcal{U}^{\text{FFN}}$).

$\mathcal{F}^{\ell,k}$ predicts $\mathcal{U}^{\ell,k}$'s normalized output $\mathbf{z}_i^{\ell,k} \triangleq \text{Norm}\left( \mathcal{U}^{\ell,k}(\mathbf{x}_i^{\ell,k}) \right)$ using *cosine similarity loss* and $L_1$ *loss*. Obviously, when the LFF outputs all zeros (i.e., when all its weights are zero), our method degenerates to greedy routing.

## 3.3 THREE-STAGE OPTIMIZATION

**Stage 1: LFF Initialization.** We commence by training the feature forecasters $\mathcal{F}^{\ell,k}$ to approximate the functional mapping of each computational unit $\mathcal{U}^{\ell,k}$. During this phase, the base LLM parameters remain *frozen*, preserving the original feature distributions. For each unit $(\ell, k)$, we minimize the forecasting loss $\mathcal{L}_{\text{fit}}^{\ell,k}$ using feature pairs $\{(\mathbf{x}_i^{\ell,k}, \mathbf{z}_i^{\ell,k})\}_{i=1}^N$ extracted from a random-selected subset (2,000 samples) of the training corpus.

This decoupled training paradigm (as shown in Figure 2(d)) admits two significant advantages:

1. *Architectural Independence*: Each $\mathcal{F}^{\ell,k}$ learns a *local approximation* of $\mathcal{U}^{\ell,k}$ without gradient propagation between computational units. This isolation eliminates inter-unit dependencies, enabling:

2. *Massive Parallelization*: Forecasters across all $L$ layers and $k \in \{\text{SA}, \text{FFN}\}$ can be trained concurrently via:

$$\underset{\theta_{\mathcal{F}^{\ell,k}}}{\text{minimize}} \mathbb{F}_{(\mathbf{x},\mathbf{z}) \sim \mathcal{D}} \left[ \mathcal{L}_{\text{fit}}^{\ell,k} \left( \mathcal{F}^{\ell,k}(\mathbf{x}; \theta), \mathbf{z} \right) \right] \quad \forall(\ell, k)$$

where $\theta$ denotes forecaster parameters and $\mathcal{D}$ the feature dataset, ensuring computational efficiency during this stage.

Feature tensors $\mathbf{X}^{\ell,k}$ and $\mathbf{Z}^{\ell,k}$ can be precomputed offline, circumventing GPU memory bottlenecks associated with full-model activations. For LLaMA3.1-8B (with 64 LFF), this stage completes in less than 5 minutes on a single NVIDIA RTX 6000 Ada GPU (48GB VRAM).

**Stage 2: Router Training.** Jointly train routers $\{\mathcal{R}^{\ell,k}\}$ with LLM and forecasters frozen. Router architecture:

$$\mathcal{R}^{\ell,k}(\mathbf{x}) = \text{Linear}_{\lfloor d_1 \rfloor \to 2} \left( \text{ReLU} \left( \text{Linear}_{d \to \lfloor d_1 \rfloor} (\mathbf{x}) \right) \right) \tag{7}$$

Although the LFF is already sufficiently lightweight, to ensure a fair comparison with baseline methods, we compromise on the parameter configuration of the router to match the parameter count of SkipGPT. Specifically, the intermediate dimension ($d_1$) of our router is 200 lower than that of SkipGPT. For instance, in the case of the Llama3.1-8B model, SkipGPT adopts an intermediate dimension of $4096/4 = 1024$, while we use $4096/4 - 200 = 824$. As a result, SkipGPT introduces a total of 268.56M parameters (routers), whereas we introduce 268.54M parameters (routers + LFF).

Composite loss integrates:

$$\mathcal{L}_{\text{route}} = \mathcal{L}_{\text{LM}} + \lambda_1 \mathcal{L}_{\text{sparse}} = \mathcal{L}_{\text{LM}} + ||\rho - S_{\text{target}}||_1 \tag{8}$$

where $\mathcal{L}_{\text{LM}}$ is the language modeling loss, $\rho$ is global compute fraction (Eq. 5) and $\lambda_1 = 8.0$ balances two objective.

**Stage 3: Parameter-Efficient Fine-tuning.** Inject LoRA adapters into attention projections $(\mathbf{W}_Q, \mathbf{W}_K, \mathbf{W}_V)$ and FFN gates:

$$\mathbf{W} \leftarrow \mathbf{W} + \mathbf{AB}, \quad \mathbf{A} \in \mathbb{R}^{d \times r_{\text{LoRA}}}, \mathbf{B} \in \mathbb{R}^{r_{\text{LoRA}} \times d}, \ r_{\text{LoRA}} = 16 \tag{9}$$

Minimize $\mathcal{L}_{\text{LM}}$ with routers/LFF frozen. This step can further recover performance.

## 4 EXPERIMENTS

### 4.1 EXPERIMENTAL SETUP

Our experimental configuration aligns with SkipGPT's setting with the following specifications:

**Models.** We validate the proposed method on the open-source Llama (Grattafiori et al., 2024) model with different scales, i.e. 3B and 8B.

**Data.** The RedPajama-Data-1T-Sample (Weber et al., 2024) corpus is utilized for both calibration and training.

**Evaluation Benchmarks.** Performance is assessed on:

- *Reasoning Tasks:* Accuracy on BoolQ (Clark et al., 2019), PIQA (Bisk et al., 2020), HellaSwag (Zellers et al., 2019), Winogrande (Sakaguchi et al., 2021), ARC-E/C (Clark et al., 2018), and OBQA (Mihaylov et al., 2018) via lm-evaluation-harness (Gao et al., 2024).
- *Perplexity:* Perplexity (PPL) on WikiText-2 (Merity et al., 2017).

**Baseline Methods.** We selected state-of-the-art static pruning and dynamic computation allocation methods for comparison, with detailed descriptions in section A.3.

### 4.2 REPRODUCIBILITY STATEMENT

Please refer to section A.2 in Appendix for detailed training/evaluating settings, and we will open-source our code upon receipt.

### 4.3 RESULTS

We conduct extensive experiments to evaluate the proposed informed routing paradigm with LFF. Our results demonstrate its advantages in training stability, efficiency, and performance preservation compared to the traditional greedy routing approaches and static compressing methods. Notably, except for SkipGPT and LFF (ours), all methods report results from LoRA finetuned models. To further demonstrate the effectiveness of our method, we report two-phase results (router training + LoRA finetune) for SkipGPT and LFF, since SkipGPT essentially serves as an ablation experiment for the informed routing component in our method.

**Inconsistency Between Language Modeling and Reasoning** Experimental results reveal an inconsistency between compressed models' language modeling (LM) capability and their reasoning performance. As shown in Table 1, while perplexity (PPL) trends generally align with reasoning accuracy, certain methods deviate. For instance, SliceGPT at 25% sparsity ranks second in PPL but drops to sixth in average reasoning accuracy. Similarly, at 40% sparsity, LFF-Router achieves better PPL than SkipGPT-LoRA yet shows a 3% drop in reasoning. These indicate that LM loss alone may not fully reflect a compressed model's reasoning ability, underscoring the need for evaluation across diverse task-specific benchmarks.

**Training Stability and Efficiency Gains of LFF Initialization** The proposed informed routing approach significantly improves training stability and efficiency by initializing the router with a pre-fit LFF. This leads to faster and smoother router convergence. Intuitively, after router training, at 25% sparsity, LFF-Router reduces PPL by 11 points compared to SkipGPT-Router; at 40% sparsity, the reduction reaches 40 points. Notably, LFF-Router at 25% sparsity outperforms fully fine-tuned SkipGPT-LoRA while saving over 50% training time (details can be find in section A.2).

Table 1: Performance comparison of different pruning methods on reasoning and language modeling tasks at sparsity levels of 25% and 40%. For reasoning tasks, we report accuracy (%); higher is better. The average (AVG) accuracy across all reasoning tasks is included. For Wikitext-2 (WT2), we report perplexity (PPL); lower is better. The best results under each sparsity level are highlighted in **bold** and the second best are underlined.

(a) Sparsity = 25%

| Method | Reasoning (Acc. ↑) | | | | | | | | WT2 (PPL ↓) |
|---|---|---|---|---|---|---|---|---|---|
| | BoolQ | OBQA | PIQA | WinoG. | Hella. | ARC-C | ARC-E | AVG | |
| Dense | 82.14 | 44.6 | 81.07 | 77.43 | 81.89 | 57.68 | 84.81 | 72.80 | 7.33 |
| *Static* | | | | | | | | | |
| SliceGPT | **72.39** | 34.4 | 66.7 | 61.56 | 56.96 | 31.48 | 50.08 | 53.37 | 9.22 |
| Shortened-llama | 71.19 | 37.4 | 73.72 | **71.82** | 69.56 | 44.45 | 66.88 | 62.15 | 10.32 |
| ShortGPT | 72.05 | 38.4 | 73.94 | 70.96 | 69.23 | 43.86 | 68.01 | 62.35 | 11.13 |
| *Dynamic* | | | | | | | | | |
| MoD | 50.28 | 31.6 | 64.25 | 52.41 | 50.44 | 28.24 | 37.67 | 44.98 | 34.21 |
| D-LLM | 50.36 | 30.2 | 57.4 | 52.49 | 37.64 | 28.16 | 37.12 | 41.91 | 40.12 |
| SkipGPT-Router | 54.13 | 27.6 | 53.92 | 54.46 | 60.92 | 39.25 | 68.31 | 51.23 | 20.76 |
| LFF-Router (ours) | 71.19 | 40.80 | 74.97 | 63.69 | 73.35 | 49.23 | 79.08 | 64.62 | 9.43 |
| SkipGPT-Lora | 70.67 | 29.60 | 56.96 | 62.83 | 74.22 | 49.91 | 78.79 | 60.43 | 10.53 |
| LFF-Lora (ours) | 71.93 | **41.80** | **76.82** | 65.19 | **76.54** | **51.45** | 79.38 | **66.16** | **8.91** |

(b) Sparsity = 40%

| Method | Reasoning (Acc. ↑) | | | | | | | | WT2 (PPL ↓) |
|---|---|---|---|---|---|---|---|---|---|
| | BoolQ | OBQA | PIQA | WinoG. | Hella. | ARC-C | ARC-E | AVG | |
| Dense | 82.14 | 44.6 | 81.07 | 77.43 | 81.89 | 57.68 | 84.81 | 72.80 | 7.33 |
| *Static* | | | | | | | | | |
| SliceGPT | **67.52** | 28.2 | 60.61 | 55.41 | 44.15 | 25.34 | 40.7 | 45.99 | 14.87 |
| Shortened-llama | 65.02 | 32.4 | 68.01 | 64.64 | 57.55 | 33.02 | 53.11 | 53.39 | 17.22 |
| ShortGPT | 65.38 | 32.0 | 68.61 | **67.32** | 58.43 | 35.32 | 53.37 | 54.35 | 18.35 |
| *Dynamic* | | | | | | | | | |
| MoD | 50.28 | 33.0 | 65.56 | 51.38 | 54.01 | 30.2 | 38.09 | 46.07 | 40.42 |
| D-LLM | 50.00 | 31.8 | 58.54 | 51.78 | 48.3 | 26.88 | 44.82 | 44.59 | 52.78 |
| SkipGPT-Router | 53.82 | 31.8 | 60.23 | 54.22 | 46.02 | 28.75 | 52.44 | 46.75 | 53.94 |
| LFF-Router (ours) | 64.43 | 36.0 | 71.87 | 52.17 | 59.95 | 37.71 | 69.99 | 56.02 | 13.87 |
| SkipGPT-LoRA | 66.57 | 37.6 | 70.78 | 56.75 | 65.17 | 42.66 | 72.39 | 58.85 | 14.35 |
| LFF-LoRA (ours) | 65.99 | 38.0 | **73.39** | 58.8 | **69.45** | 43.6 | 72.43 | 60.24 | **11.11** |

**Superiority After Fine-tuning and Underlying Mechanisms** After LoRA fine-tuning, our method outperforms SkipGPT in 15 out of 16 tasks. We attribute this to two factors: **(1)** The LFF better preserves the original feature distribution by approximating the layer transformation instead of discarding tokens. Features processed by LFF show higher cosine similarity and lower L1 loss (0.16 vs. 0.52), providing a warmer start for fine-tuning. **(2)** Pre-fitting the LFF enables the router to prioritize tokens with high recoverability—those predictable by a simple network—leading to a healthier model structure and better parameter recovery during LoRA fine-tuning.

## 4.4 FURTHER ANALYSIS

Table 2: Reduction ratio between Attention and MLP modules at different global sparsity levels.

| Method | 25% Sparsity | | 40% Sparsity | | 70% Sparsity | |
|---|---|---|---|---|---|---|
| | Attention | MLP | Attention | MLP | Attention | MLP |
| SkipGPT | 58.0% | 42.0% | 57.8% | 42.2% | 56.4% | 43.6% |
| LFF | 71.4% | 28.6% | 67.2% | 32.8% | 66.2% | 33.8% |

**Analysis of Router Behavior** As shown in Table 2, our method consistently select more tokens from self-attention modules than the SkipGPT baseline, across all sparsity levels. This supports prior findings He et al. (2024) that self-attention is more redundant than FFN blocks. The success of our lightweight, linear LFF in predicting attention outputs suggests that many token transformations in self-attention are approximable by simple linear operations. We term this property **linear simplicity**. Our router, preconditioned by the LFF, learns to identify such tokens, leading to a more explainable sparsity profile.

**Balanced computation between attention and FFN**    A potential point of discussion is our treatment of self-attention and FFN modules as equally valid candidates for computation reduction, a design choice inherited from SkipGPT. While self-attention contains fewer parameters, its computational complexity scales quadratically with sequence length, often making it the dominant computational bottleneck in modern long-context large language models. To preemptively address any concern that a direct comparison might be unfair, we conducted a rigorous ablation study. In this experiment, we independently controlled and enforced identical sparsity levels for each module type—self-attention and FFN—across all layers. This ensures a perfectly equitable comparison of the routing strategies' efficiency on a per-module-class basis. The results, showed in Table 3, demonstrate that our informed routing paradigm consistently achieves superior performance compared to the greedy routing baseline under these controlled sparsity conditions. This finding robustly confirms that the performance gain of our method is not an artifact of an imbalanced reduction strategy but is intrinsically linked to its preservation of features distributions, validating our core hypothesis.

Table 3: Performance of SkipGPT-balance and LFF-balance methods under balanced computation reduction setting. All results are reported via LoRA-finetuned model at 25% sparsity.

| Method | Reasoning (Acc. ↑) | | | | | | | | WT2 (PPL ↓) |
|---|---|---|---|---|---|---|---|---|---|
| | BoolQ | OBQA | PIQA | WinoG. | Hella. | ARC-C | ARC-E | AVG | |
| SkipGPT-balance | 67.77 | 39.60 | 62.08 | 60.93 | 71.33 | 40.44 | 66.84 | 58.42 | 11.40 |
| LFF-balance | **70.54** | **39.80** | **74.71** | **63.62** | **73.16** | **47.24** | **75.38** | **63.49** | **9.49** |

Table 4: Performance of SkipGPT-LoRA and LFF-Router across different sparsity. Even without the final fine-tuning stage, LFF-Router can already surpass SkipGPT with both router and LLM fine-tuning.

| Method | 25% Sparsity | | 40% Sparsity | | 70% Sparsity | |
|---|---|---|---|---|---|---|
| | Val. Loss↓ | PPL ↓ | Val. Loss↓ | PPL↓ | Val. Loss↓ | PPL↓ |
| SkipGPT-LoRA | 2.38 | 9.61 | 2.74 | 14.34 | **3.72** | **52.75** |
| LFF-Router | **2.36** | **9.46** | **2.57** | **12.89** | 4.08 | 88.17 |

**Can informed routing increase the upper limit of computational reduction?**    We investigate whether the proposed *informed routing* approach can elevate the upper bound of computational reduction. Empirical results suggest otherwise. As shown in Table 4, at 25% sparsity, LFF-Router *(require only LFF initialization and router training, without LoRA finetuning)* surpasses the full training of SkipGPT-LoRA, indicating effective feature forecasting. However, at 40% sparsity, LFF-Router fails to outperform SkipGPT-LoRA in reasoning tasks, revealing its forecasting limits. At 70% sparsity, the gap widens substantially in language modeling, confirming that LFF's capacity is exceeded. Thus, while effective at moderate sparsity, informed routing does not extend the ultimate computation reduction boundary.

Table 5: Performance comparison on Llama3.2-3B with 25% sparsity. Accuracies (%) on reasoning tasks; perplexity (PPL) on WikiText-2.

| Method | Reasoning (Acc. ↑) | | | | | | | | WT2 (PPL ↓) |
|---|---|---|---|---|---|---|---|---|---|
| | BoolQ | OBQA | PIQA | WinoG. | Hella. | ARC-C | ARC-E | AVG | |
| Dense | 73.03 | 43.40 | 77.58 | 72.22 | 76.41 | 50.85 | 79.17 | 67.52 | 9.27 |
| SkipGPT-Router | 47.65 | 34.40 | 61.15 | 54.14 | 51.99 | 26.62 | 39.56 | 45.07 | 36.50 |
| LFF-Router (ours) | 61.74 | 34.80 | 68.28 | 58.33 | 62.36 | 40.53 | 71.04 | 56.73 | 12.19 |
| SkipGPT-LoRA | 62.81 | 36.60 | 66.49 | **59.19** | 64.52 | 39.51 | 70.71 | 57.12 | 14.82 |
| LFF-LoRA (ours) | **62.87** | **37.20** | **69.04** | 59.04 | **66.14** | **41.81** | **73.06** | **58.45** | **11.46** |

**Generalization on Llama-3B**    To validate the generalization across model scales, we evaluate our method on the Llama3.2-3B model. As shown in Table 5, the results align with those from the 8B model, substantiating our approach's efficacy. At 25% sparsity, LFF-Router significantly outperforms SkipGPT-Router, improving the average accuracy on reasoning tasks by 11% and reducing language modeling perplexity by 24. Moreover, LFF-Router matches the performance of SkipGPT-LoRA while saving over 50% in training time. After LoRA fine-tuning, LFF-LoRA achieves superior performance on 8 out of 9 tasks, confirming the advantage of informed routing. An key finding is

that the performance degradation is more pronounced on the 3B model, indicating its lower intrinsic redundancy and higher sensitivity to computation reduction.

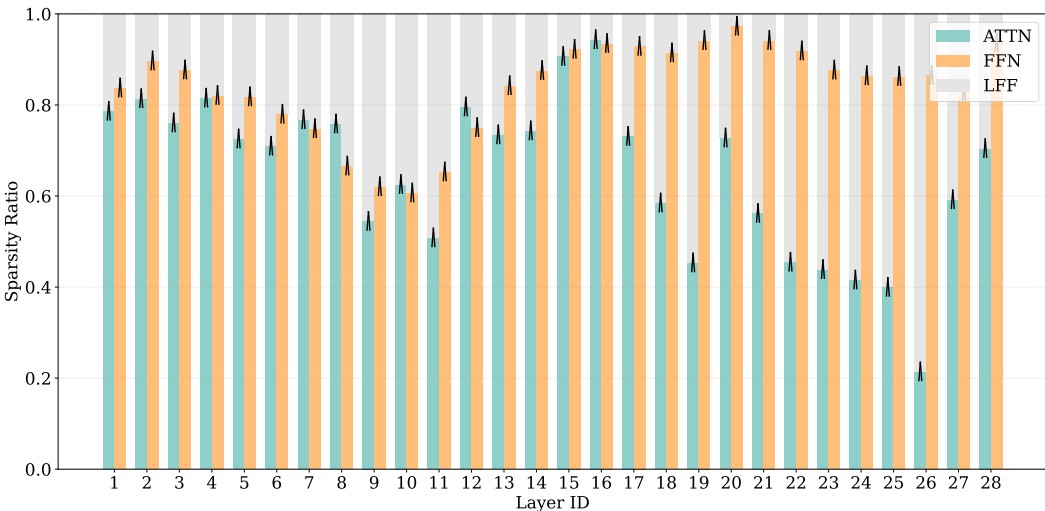

Figure 3: Layer-wise Token Allocation. The hatched area represents the proportion of tokens processed by the efficient LFF branch and the colored areas show the tokens retained for full computation in the Attention (green) and FFN (orange) modules.

**Allocation Visualization for Attention and FFN Modules** This section presents an intuitive visualization of token allocation by routers across Attention (ATTN) and Feed-Forward Network (FFN) modules within the Llama-3B model. We track and record the layer-wise allocation information for each batch on the validation set, and then present the average value across all batches.

As shown in Figure 3, which details the allocation across 28 layers, the routing mechanism intelligently distributes input tokens, creating a dynamic computational sparsity pattern. A key observation is that the proportion of tokens directed to the LFF branch (hatched area) varies significantly across layers (also between attention and FFN modules), suggesting that the router adapts its filtering strategy based on the hierarchical processing needs of the network. This strategic allocation preserves critical tokens for the full computational pipeline while efficiently processing others, effectively reducing the overall computational overhead without compromising performance.

## 5 CONCLUSION

In this work, we identified fundamental limitations in the established greedy routing paradigm for dynamic computation reduction in large language models: its reactive nature leads to irreversible information loss and its token selection criterion is inherently short-sighted. In response, we proposed a paradigm-shifting alternative, informed routing, which introduces Lightweight Feature Forecasters to fit inter-layer transformations before routing decisions are made. Our approach offers key advantages: First, LFFs approximate skipped tokens, reducing feature shift and improving initial stability with less performance drop. Second, LFF forecasting error gives the router a recoverability-based importance measure, enabling smarter retention of hard-to-predict tokens. Third, our method consistently outperform greedy routing methods on both unbalanced and balanced reduction setting. Finally, we show self-attention's redundancy stems from linearly approximable transformations. Despite these advancements, our exploration of extreme sparsity levels (e.g., 70%) reveals that the upper limit of dynamic computation allocation is ultimately governed by the complexity of the underlying transformations, which a simple LFF cannot fully capture. This presents an exciting avenue for future work, which could explore more sophisticated yet efficient forecasters or hybrid strategies.

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

# A APPENDIX

## A.1 STATEMENT ON LARGE LANGUAGE MODEL USAGE

The authors use Deepseek-r1(DeepSeek-AI et al., 2025) solely for text refinement, including grammar checking, polishing, and condensing sections to meet length constraints.

## A.2 EXPERIMENT DETAILS

**Training** Hyper-parameters differ across stages, all stages adopt the same AdamW optimizer (Loshchilov & Hutter, 2017) ($\beta_1 = 0.9$, $\beta_2 = 0.95$).

- *Forecaster Initialization.* Constant learning rate ($1e^{-3}$), training steps (2000), batchsize (8).
- *Router Tuning:* Constant learning rate ($2 \times 10^{-3}$), training steps (2000), batchsize (16).
- *LoRA Tuning:* Cosine annealing learning rate ($2 \times 10^{-3}$), training steps (2000) with warmup steps (200), batchsize (16).

It is worth emphasizing that in our experiments, we found that computing the router and LFF after normalization (e.g., RMSNorm in llama) can improve training stability—especially when the LFF involves some non-linear activations (e.g., Swish). Therefore, we strongly recommend computing the router and LFF after normalization, which is exactly the approach we adopted in our experiments (to both SkipGPT and LFF).

All experiments were conducted on a single NVIDIA RTX 6000 GPU with 48 GB VRAM. The time consumption of the three experimental phases is summarized in Table 6.

| Experimental Phase | Time Consumption |
|---|---|
| LFF Initialization | 5 minutes |
| Router Training | 3 hours |
| LoRA Finetuning | 4 hours |

Table 6: Time consumption of different experimental phases.

**Evaluating** We use lm-eval (Gao et al., 2024) for all evaluation tasks with version 0.4.9. And followed SkipGPT (Zhao et al., 2025), tasks are evaluated with different few-shot contexts, details are listed in Table 7.

| Task Name | Number of Few-shot Examples | Evaluation Metric |
|---|---|---|
| OpenBookQA | 0 | acc_norm |
| Winogrande | 5 | acc |
| PIQA | 0 | acc |
| HellaSwag | 10 | acc_norm |
| BoolQ | 0 | acc |
| ARC-Easy | 25 | acc_norm |
| ARC-Challenge | 25 | acc_norm |
| WikiText2 | 0 | word_perplexity |

Table 7: Configuration of few-shot examples and evaluation metrics for different tasks.

### A.3 COMPARISON METHODS

To provide a comprehensive evaluation, the proposed method is compared with several state-of-the-art approaches in static model compression and dynamic computation allocation.

- **SliceGPT** (Ashkboos et al., 2024): This method applies Principal Component Analysis (PCA) on orthogonally transformed parameters to remove entire rows and columns, achieving static parameter pruning. It results in a uniformly smaller model by permanently removing fixed structural components.

- **Shortened-llama** (Kim et al., 2024): This approach focuses on depth pruning by removing consecutive layers in LLMs to create a smaller model. It demonstrates that reducing model depth can be an efficient strategy for LLM inference.

- **ShortGPT** (Men et al., 2024): Leveraging Block Influence (BI), ShortGPT quantitatively estimates the importance of layers to prune less critical ones. This method is a static layer-pruning technique that aims to reduce model capacity.

- **Mixture-of-Depths (MoD)** (Raposo et al., 2024): MoD is a dynamic computation allocation method that enforces a fixed sparsity ratio per layer block. It employs a greedy routing paradigm where routers decide to execute or skip computational units for tokens.

- **D-LLM** (Jiang et al., 2024): This method introduces global adaptive sparsity, dynamically allocating computation across layers based on input characteristics. It refines dynamic computation by allowing more flexible computation paths across the model.

- **SkipGPT** (Zhao et al., 2025): A prominent dynamic computation allocation baseline, SkipGPT further refines granularity by decoupling attention and MLP operations within each layer. It uses separate routers to independently skip sub-modules, operating under the greedy routing paradigm.

### A.4 ANALYSIS ON KEY-VALUE CACHE REDUCTION

While the primary design of SkipGPT and our method focuses on computational reduction, the memory footprint of the Key-Value (KV) Cache remains a critical bottleneck in autoregressive transformer inference. To directly target KV cache reduction, we adopt an aggressive strategy following (Jiang et al., 2024): an additional masking mechanism is applied to prevent normal tokens from attending to any skipped (or route to LFF branch) tokens in the sequence, to simulate that removing the selected tokens' key-value pairs from the cache.

The performance impact of this operation is non-negligible, as it alters the model's fundamental attention pattern. Our experiments (TABLE 8) confirm that enforcing this strict KV cache reduction leads to a predictable degradation in model quality. The training convergence loss increases by 0.19, and the perplexity on WikiText2 rises by 2.3 points compared to the standard pruning setup which retains the full cache. This decline underscores a direct trade-off between memory compression and model fidelity; removing information from the attention context inevitably impairs the model's representational capacity.

Table 8: Performance of models with/without KV reduction at 25% sparsity

| Method | Validation Loss | Final PPL ↓ |
|---|---|---|
| w.o KVR | 2.30 | 8.93 |
| w. KVR | 2.49 | 11.21 |

Table 9: Performance of models with LFFs of different intermediate dimensions at 25% sparsity. Performance saturates beyond a dimension of 50.

| Inter. Dim | 10 | 50 | 100 | 500 | 1000 |
|---|---|---|---|---|---|
| Params. (M: $10^6$) | $\sim$0.082 | $\sim$0.41 | $\sim$0.82 | $\sim$4.10 | $\sim$8.20 |
| Final PPL ↓ | 15.9 | 9.3 | 8.9 | 8.9 | 8.8 |

## A.5 THE SELECTION OF LIGHTWEIGHT FEATURE FORECASTER

The design of the LFF is central to our *informed routing* paradigm. We use a two-layer linear network motivated by two factors: **first**, its simplicity helps validate our core hypothesis—that forecasting before routing stabilizes the training process—without obscuring the gains; **second**, it is highly parameter-efficient, avoiding computational overhead that could undermine acceleration. An ablation on the LFF's intermediate dimension (Table 9) shows performance saturates beyond a dimension of 50, suggesting only a limited subset of token transformations are simple enough to be captured linearly.

Naturally, a more complex forecaster (e.g., an architecture identical to the original model block represents the theoretical upper bound of performance) could achieve better accuracy but offers less speedup, defeating the purpose of inference acceleration. Thus, the LFF lies on a Pareto frontier between performance and efficiency. Our framework allows users to configure its complexity based on specific accuracy-speed trade-offs, ensuring adaptability across scenarios.

