# OpenReview forum: "Bring Future Vision: Dynamic Computation Allocation Guided by Lightweight Feature Forecaster"
_ICLR.cc/2026/Conference — Submitted to ICLR 2026_

### Official Review · Reviewer_17Wo · 2025-10-24

**Soundness:** 3
**Presentation:** 3
**Contribution:** 2
**Rating:** 2
**Confidence:** 5

**Summary:**

This paper proposes Informed Routing, a new dynamic computation allocation framework for large language models that aims to improve both efficiency and performance in token-wise pruning. Existing DCA approaches rely on greedy routing, where routers make binary “execute-or-skip” decisions that can cause irreversible information loss.

The proposed method introduces a Lightweight Feature Forecaster, a small, efficient module that approximates the output of transformer submodules. Instead of skipping tokens outright, the router decides whether to process them normally or approximate them via LFF, based on their recoverability.

This shift from “skip” to “approximate” allows smoother performance degradation under sparsity. Experiments on LLaMA-3B and 8B models show that Informed Routing achieves better accuracy–perplexity trade-offs than baselines like SkipGPT and static pruning methods, especially at 25–40% sparsity.

**Strengths:**

1. The proposed Informed Routing paradigm and LFF module offer a new way to dynamically adjust computation without severe performance loss. Results on reasoning and language modeling tasks show clear gains over both static pruning and greedy dynamic baselines, proving its effectiveness.
2. The experiments are comprehensive and systematic, covering both static and dynamic pruning comparisons across sparsity levels, model sizes (3B and 8B), and diverse benchmarks. The ablation studies further clarify how different LFF sizes, sparsity ratios, and module balances influence performance.

**Weaknesses:**

1. Although the framework improves theoretical performance under sparsity, its real-world efficiency gain is questionable. In large-batch inference, skipped or approximated tokens must wait for dense batches, making parallel execution inefficient. Thus, the approach may not outperform dense models in actual deployment scenarios.
2. Pruning inevitably causes non-negligible accuracy loss. The authors’ own results (e.g., Table 4 and 5) show that while moderate sparsity (25%) retains quality, higher sparsity ratios rapidly degrade performance. This limits the usefulness of the framework in real-world LLM compression, where the efficiency–performance trade-off remains hard to beat compared to dense baselines or specialized architectures.

**Questions:**

1. Since pruning inevitably leads to information loss, how can this dynamic computation model meaningfully outperform dense models in overall efficiency once practical acceleration and batch synchronization are considered?
2. In parallel inference scenarios, the dynamic skipping introduces heterogeneous batch latencies. How is this addressed to ensure consistent throughput? Are there strategies to mitigate the “waiting problem” of sparse batches?

---

> ### Author Response · Authors · 2025-11-24
> **Response to Reviewer 17Wo**
>
> Thank you very much for your critical feedback. We have carefully studied your review comments and note that your concerns primarily stem from the perspective that "dynamic computation allocation cannot achieve theoretical acceleration in large-batch scenarios," with the associated performance overhead appearing unacceptable. However, we believe this view might be somewhat one-sided, and we would like to provide the following clarifications:
>
> **1. Practical Application Scenarios and Existing Acceleration Evidence**
>
> The issue you raised is entirely non-existent when batch size = 1. Scenarios with a batch size of 1 hold significant practical value, such as in personal LLM assistants or LLM-based robotics. Furthermore, our results demonstrate that in long-sequence scenarios, the proposed algorithm achieves theoretical speedup even with a naive PyTorch implementation. For short-sequence scenarios, further acceleration is attainable through customized operators (please refer to **Official Comment: Wall-clock speed and the selection of LFF**).
>
> **2. Validation and Acceptance of Dynamic Computation**
>
> Dynamic computation has emerged as an active and recognized research direction, evidenced by its adoption in recent top-tier academic conferences and by fellow researchers. Representative works include MoD [Arxiv24, 167+ citations], d-LLM [NeurIPS24], and SkipGPT [ICML25].
>
> * MoD: Mixture-of-Depths: Dynamically allocating compute in transformer-based language models
> * d-LLM: D-LLM: A Token Adaptive Computing Resource Allocation Strategy for Large Language Models
> * SkipGPT: SkipGPT: Dynamic Layer Pruning Reinvented with Token Awareness and Module Decoupling
>
> **3. Potential Solutions for Large-Batch Scenarios**
>
> Even in large-batch scenarios with varying request lengths (e.g., sequences exhibiting different sparsity levels), we can employ techniques to reduce redundant computation and achieve tangible speedup. The key lies in minimizing invalid padding. Several promising technical approaches include:
>
> *   **Prepacking** (Implemented in major inference frameworks like vLLM): This method packs all input tokens into a single contiguous sequence (effectively treating the batch as batch size = 1), primarily targeting scenarios with prompts of varying lengths. For dynamic computation allocation, after the router makes decisions at each layer, the tokens within the batch effectively resemble prompts of different lengths. Prepacking would simply need to be applied once per layer following the router's decision.
>
> *   **Chunked Prefill** (Implemented in vLLM): During the prefill phase, the framework processes long sequences in segments (chunks). This means that shorter sequences within the batch are absent in later chunks, capping the maximum computational waste to within a single chunk. By keeping the chunk size sufficiently small, computational acceleration can be achieved. Similar to prepacking, this method would require executing the chunking process once per layer after the router's decision. While introducing some overhead, this cost is likely negligible compared to the substantial computation of the large model itself, suggesting remaining potential for net acceleration.

---

> ### Comment · Reviewer_17Wo · 2025-11-25
>
> Dear authors:
>
> Thanks for your feedback. The major weakness not only lies on the limited acceleartion in large batches, but also on the limited sparsity ratio on quality-retained settings.
>
> While we agree about the fact that the proposed method is difficult to tackle with large batches, the practical value of single-batch acceleration is a subjective judgment. The statements about ideal application and related works are not persuasive to me. If the proposed method only works in specific fields, the paper may be not appropriate for top-tier machine learning conferences.
>
> Since there is no misunderstanding about the paper itself, I will keep my score if no significant change happens.

---

### Official Review · Reviewer_uWVk · 2025-10-27

**Soundness:** 3
**Presentation:** 3
**Contribution:** 3
**Rating:** 4
**Confidence:** 2

**Summary:**

This paper proposes the Informed Routing paradigm, where a lightweight network called LFF (Lightweight Feature Forecaster) predicts the output of a computation block before the router decides whether to execute or approximate it. This replaces “skipping” with “approximating”,  mitigating information loss and improving routing decisions. Experiments on LLaMA-family models  for language modeling  and reasoning tasks show that the proposed method achieves better performance and training efficiency (reporting over 50% training time savings) than greedy routing baselines.

**Strengths:**

1. Novel and intuitive motivation: The idea of using predictability as a routing criterion directly addresses the weaknesses of greedy routing and is well motivated with quantitative metrics (cosine similarity and L1 error).


2. Lightweight design: The LFF module is compact, adding only about 0.02% extra parameters (≈0.82M), achieving a good trade-off between model capacity and efficiency.

3. Comprehensive experiments on small and mid-sized models: The method is evaluated on 3B and 8B models, with multiple sparsity ratios (25/40/70%) and ablations (balanced/unbalanced routing). Results demonstrate consistent improvements across settings.

**Weaknesses:**

1. Code release and reproducibility: The paper states “we will open-source our code upon receipt,” but no repository or scripts are currently provided. The authors should release code and scripts for LFF initialization, router training, LoRA fine-tuning, and inference benchmarks (including environment setup and seeds).

2. Lack of evaluation on larger models: Experiments are limited to 3B and 8B models. It is unclear whether LFF remains effective for larger models such as 13B, 30B, or 70B, where attention/FFN balance and KV-cache behavior differ.

3. Missing wall-clock speed and latency benchmarks: The paper reports training time savings (“>50% faster”) but does not provide inference-time throughput or latency results, which are critical to validate claims of deployment efficiency.

**Questions:**

See weaknesses.

---

> ### Author Response · Authors · 2025-11-24
> **Response to Reviewer uWVk**
>
> Thank you very much for your valuable feedback and thoughtful review. I truly appreciate your time and insights. Below, I will address each of your points in sequence.
>
> **1. Comment: Code release and reproducibility...**
>
> We sincerely apologize for not providing the repository link in the submission. This was primarily due to the ICLR double-blind policy requirements. Our research group maintains a consistent practice of open-sourcing code, which unfortunately makes author information easily traceable through our existing public repositories. We will be happy to release the complete code repository immediately upon acceptance.
>
> **2. Comment: Lack of evaluation on larger models...**
>
> We acknowledge the reviewer's interest in scaling our method to larger models. Currently, **we lack the computational resources** to conduct extensive experiments on models beyond the scale used in our paper (e.g., 3b,8b). However, based on our methodology's design:
> - The router and linear networks are **lightweight add-ons** trained separately per layer, so the overhead should remain minimal even for larger models.
> - The core idea—**dynamic computation allocation via a router**—is architecture-agnostic and theoretically scalable. Prior work in dynamic computation has shown promise on larger models (e.g., in 13b models [SkipGPT, ICML2025]), suggesting our approach could generalize.
>
> **3. Comment: Missing wall-clock speed and latency benchmarks...**
>
> We fully acknowledge the importance of wall-clock speed measurements and have accordingly reported acceleration results using both pure Python implementations and customized operators (please refer to **Official Comment: Wall-clock speed and the selection of LFF**). To summarize: our naive Python implementation demonstrates that the proposed method achieves theoretical speedups for long sequences under high sparsity conditions. For scenarios involving shorter sequences or lower sparsity levels, we have also shown that customized operators can significantly improve performance.

---

### Official Review · Reviewer_FY6Y · 2025-10-29

**Soundness:** 3
**Presentation:** 3
**Contribution:** 3
**Rating:** 6
**Confidence:** 4

**Summary:**

This paper identifies and addresses key limitations in existing dynamic computation allocation (DCA) methods for large language models (LLMs). The authors argue that the prevalent "greedy routing" paradigm, which forces a binary "execute-or-skip" decision for each token, leads to irreversible information loss and suboptimal token selection based on short-sighted criteria. To overcome this, they propose a new paradigm called "informed routing," which replaces the skip option with an efficient approximation. The core of this approach is the Lightweight Feature Forecaster (LFF), a small network trained to mimic the transformation of its corresponding model component. This allows the router to base its decision on a token's "recoverability" (i.e., how well its transformation can be approximated) rather than its immediate importance. The proposed method involves a three-stage training pipeline: LFF initialization, router training, and optional LoRA fine-tuning. Extensive experiments demonstrate that this approach achieves state-of-the-art performance, significantly improves training stability, and can match or exceed fully fine-tuned baselines even without the final fine-tuning step, reducing training time by over 50%.

**Strengths:**

- The paper introduces a novel and significant paradigm shift from "greedy routing" to "informed routing". The concept of replacing an information-destroying "skip" action with an information-preserving "approximate" action is a creative and powerful idea. Using the LFF to assess a token's recoverability is an original criterion for routing decisions that directly addresses the core weaknesses of prior work.
- The empirical evaluation is comprehensive and of high quality. The authors validate their method on multiple model scales (3B and 8B Llama models) and across a diverse set of reasoning and language modeling benchmarks. The inclusion of strong static and dynamic baselines, along with insightful ablation studies like the balanced computation experiment, provides robust evidence for the method's effectiveness.
- The paper is exceptionally clear and well-written. The motivation is well-established by clearly diagnosing the "All-or-Nothing Dilemma" and "Short-Sighted Token Selection" problems of existing methods. The proposed architecture and training pipeline are explained logically and are well-supported by clear diagrams (e.g., Figure 2), making the work easy to understand and reproduce.
- The work carries significant practical implications. The finding that the method, after only router training, can outperform a fully fine-tuned baseline (SkipGPT-LORA) while saving over 50% of the training time is a major advantage. This highlights the efficiency and stability gains from the LFF pre-fitting stage, making advanced model compression more accessible.

**Weaknesses:**

- The exploration of the Lightweight Feature Forecaster (LFF) architecture is somewhat limited. The paper defaults to a simple two-layer linear network, which, while efficient, may not capture the full potential of the informed routing paradigm. The ablation study shows performance saturating with a small intermediate dimension, suggesting that this simple linear model hits a performance ceiling quickly. An investigation into slightly more complex, non-linear LFFs could provide valuable insights into the trade-off between forecaster capacity and overall model performance, especially at higher sparsity levels where the current LFF struggles.
- The method's contribution to Key-Value (KV) cache reduction feels underdeveloped. The authors adopt a straightforward masking strategy that predictably degrades performance and is not deeply integrated with the core LFF mechanism. Since the LFF approximates a token's output rather than eliminating it, the token's key and value are still required for subsequent layers, meaning the approach does not inherently reduce the KV cache. A more novel approach that leverages the LFF's predictive capabilities to perhaps generate approximate KV pairs could have been a more impactful contribution.
- While the paper demonstrates strong performance at moderate sparsity (25-40%), it also concedes that the approach does not extend the ultimate upper bound of computation reduction, with performance dropping off significantly at 70% sparsity. This positions the method as an improvement upon existing techniques rather than a complete breakthrough. A discussion on potential hybrid strategies or adaptive LFFs that become more powerful as sparsity increases could have strengthened the paper's forward-looking perspective.

**Questions:**

- The LFF initialization uses a fixed, relatively small dataset of 2,000 samples. How sensitive is the LFF's approximation quality, and consequently the final model's performance, to the size and composition of this initialization dataset? Could a larger or more strategically sampled dataset allow the LFF to learn more robust transformations?
- The paper provides an interesting analysis showing that your method prunes attention modules more heavily, attributing this to their "linear simplicity". This is a fascinating claim. Could you provide more qualitative or quantitative analysis to support this? For instance, what types of tokens or linguistic phenomena are most frequently routed through the LFF in attention layers versus FFN layers?
- The use of LFFs introduces approximation errors for a subset of tokens at each layer. How do these small errors propagate and accumulate through the network? Is there a risk that the accumulated noise from multiple LFFs could negatively impact the representations of tokens that are consistently processed by the main computational units in later layers?
- The paper's evaluation focuses primarily on model quality metrics (accuracy and perplexity) rather than practical acceleration. Could the authors provide a quantitative analysis of the wall-clock inference speedup (e.g., in tokens/second or latency) for both the LFF method and the SkipGPT baseline at various sparsity levels? It would be particularly insightful to see how these realized speedups compare to the theoretical speedup suggested by the sparsity ratio, and what overheads (e.g., router logic, non-contiguous memory access) limit performance. Similarly, for the training phase, what is the acceleration per training step compared to a dense model, to isolate the computational savings from the faster convergence you already reported?

---

> ### Author Response · Authors · 2025-11-24
> **Response to Reviewer  FY6Y**
>
> Thank you very much for your detailed and constructive feedback. I truly appreciate the time and effort you've invested in reviewing our work. Below, I will address each of your points and questions one by one.
>
> **1. Comment: The exploration of the Lightweight Feature Forecaster (LFF) architecture is somewhat limited...**
>
> During the initial submission period, we actually explored simple non-linear architectures. Specifically, we experimented by adding a non-linear layer (e.g., 4096-100-ReLU-4096) after the first linear mapping. However, experimental results demonstrated that this did not yield significant improvements compared to a purely linear model, and thus it was not included in the final manuscript. Recently, we have revisited the LFF design, incorporating a 'slimmed' LLM approach—for instance, by reducing the intermediate dimension of the original LlamaMLP. Our experiments (please refer to **Official Comment: Wall-clock speed and the selection of LFF**) indicate that this design can improve overall model performance under high sparsity conditions. Admittedly, this approach introduces approximately double the parameters compared to a simple linear model. In summary, we believe the choice of LFF can be tailored to practical application needs: for lower sparsity levels, a simple linear network suffices; for higher sparsity, careful consideration of the LFF design is necessary to better approximate the original LLM tokens, and perhaps integrating Neural Architecture Search (NAS) could be a promising direction.
>
> **2. Comment: The method's contribution to Key-Value (KV) cache reduction feels underdeveloped...**
>
> In our experiments, we employed a masking strategy to simulate KV cache reduction primarily for ease of implementation within existing frameworks (like PyTorch and Transformers). Our algorithm is inherently capable of achieving actual KV cache reduction (albeit with some performance trade-off). This is because the integration of subsequent tokens does not alter the representation of preceding tokens, thereby maintaining consistent router behavior. Specifically, if a token is routed through the LFF branch at a certain layer, it will continue to bypass the original LLM attention branch in all subsequent decoding steps. Furthermore, we think the concept of simulated KV cache quite intriguing and believe it could be a valuable direction for memory-bounded decoding phase.
>
> **3. Comment: While the paper demonstrates strong performance at moderate sparsity (25-40%)...**
>
> The performance ceiling observed is primarily constrained by the current capabilities of the LFF, preventing the method from fully reaching the theoretical upper bound of dynamic computation allocation. As mentioned in our response to comment #1, we are actively exploring enhanced LFF architectures to improve overall effectiveness. Additionally, we acknowledge that any compression technique inherently has a performance ceiling—beyond which performance degrades precipitously. Drawing an analogy to natural language: while redundant text can be condensed without losing essential meaning, there exists a critical threshold below which the compressed information becomes insufficient for accurate comprehension. This fundamental limit similarly applies to token compression methods like ours.
>
> **4. Comment: The LFF initialization uses a fixed, relatively small dataset of 2,000 samples...**
>
> LFFs are trained using a local alignment strategy, meaning each LFF is tasked solely with approximating the output of its corresponding LLM module. Given their minimal parameter count, the training process for LFFs is inherently stable and straightforward. Importantly, since LFFs are simple linear networks rather than complex transformers, each token in an input sequence effectively serves as an individual training sample. Therefore, a dataset of 2000 text sequences translates to up to 2000 * 2048 (the maximum token length) training instances, which we have found to be more than sufficient. In practice, our experiments indicate that even fewer samples (around 800-1000) are often adequate for stable LFF training.

---

> > ### Author Response · Authors · 2025-11-24
> > **Response to Reviewer  FY6Y**
> >
> > (divided into two parts due to word limits)
> >
> >
> > **5. Comment: The paper provides an interesting analysis showing that your method prunes attention modules more heavily...**
> >
> > We have already compiled the relevant token routing statistics on the Wikitext2 corpus. The detailed analysis is currently ongoing, and we will promptly update the manuscript once we have concrete conclusions to report.
> >
> > **6. Comment: The use of LFFs introduces approximation errors for a subset of tokens at each layer...**
> >
> > We agree that error propagation is influenced by both the representational capacity of the LFF and the target sparsity level. When the LFF's capacity is adequate for the chosen sparsity—for instance, at 25% sparsity, where a simple LFF initialization combined with router training matches the performance of methods requiring additional LoRA fine-tuning—it suggests that the accumulated error across layers remains minimal. Furthermore, from a feature similarity perspective, using an LFF to approximate the original output results in higher feature similarity compared to alternative approaches (e.g., directly reusing features from previous layers). This indicates that the LFF plays a beneficial role in mitigating overall error propagation (achieving average token similarity of 0.94-0.97 for attention modules and 0.86-0.89 for MLP modules).
> >
> > **7. Comment: The paper's evaluation focuses primarily on model quality metrics (accuracy and perplexity) rather than practical acceleration...**
> >
> > We fully recognize the importance of demonstrating practical acceleration and have included results from both a pure Python implementation and a customized kernel implementation (please refer to **Official Comment: Wall-clock speed and the selection of LFF**). To summarize: our naive Python implementation confirms that the proposed method can achieve theoretical speedups for long sequences and high sparsity ratios. For scenarios involving shorter sequences or lower sparsity, we have also shown that customized operators can significantly enhance performance.

---

> > > ### Author Response · Authors · 2025-12-03
> > > **respond to comment 5: Visualization of Skipped Tokens from a Natural Language Perspective**
> > >
> > > Thank you for raising this point. To analyze which tokens are skipped, we sampled a subset of **~100k tokens** from **WikiText-2** (due to computational constraints) and aggregated the total skip (which means approximated by LFF for our method) counts across all layers of the model. The table below shows representative examples of most/least frequently skipped tokens and their counts (we omit zero-skipped tokens as they are overwhelming in number):
> > >
> > > | Highly Skipped Tokens | Skip Count || Rarely Skipped Tokens | Skip Count |
> > > |-------|------------|---|---|---|
> > > | ,   | 870    || more | 4
> > > | the    | 663    || ich |6
> > > | [space]     | 588    || Destiny	|7
> > > | in   | 468    || named | 7
> > > | and    | 397    || Eastern|7
> > > | @     | 300    ||ot | 7
> > > | to    | 282    ||ination|7
> > > | '     | 244    ||781	| 8
> > > | of    | 235    ||lengthy|8
> > >
> > > **Interpretation:**
> > > The model tends to skip **function words** (e.g., conjunctions like "and", prepositions like "in"), **punctuation marks** (e.g., ",", "."), and other low-information tokens. This aligns with intuition: such tokens contribute minimally to semantic meaning and can be approximated by simple linear projections without significant loss. Our method's router learns to allocate complex computations to content-rich tokens (e.g., nouns, verbs).

---

### Official Review · Reviewer_gfbw · 2025-10-30

**Soundness:** 3
**Presentation:** 3
**Contribution:** 3
**Rating:** 6
**Confidence:** 2

**Summary:**

This paper proposes Informed Routing for dynamic computation allocation in LLMs. Instead of the traditional greedy execute-or-skip decision, the authors introduce a new option: execute-or-approximate via Lightweight Feature Forecasters (LFFs), which are small bottleneck networks trained to mimic the output of each compute unit (self-attention or FFN) before routing decisions are made. The method is trained in three stages: (1) prefit LFFs to each unit, (2) train routers (with Gumbel-Softmax) to choose between the original unit and its LFF under a global sparsity target, and (3) optional LoRA fine-tuning. Experiments on LLaMA-3B/8B report lower perplexity and higher reasoning accuracy than strong static and dynamic baselines (e.g., SkipGPT), especially at 25% sparsity, with >50% training-time savings for the router+LoRA pipeline. Analyses indicate attention transformations are often linearly simple and thus more predictable by LFFs, while very high sparsity (e.g., 70%) exposes forecasting limits.

**Strengths:**

1. Clear motivation: The authors motivates the invention of LFFs well by pointing out that current DCA method adopt a all-or-nothing approach which has room for improvements. This gives the router a recoverability signal in terms of how well a token's transformation can be forecast.

2. Clean methodology: The three stage pipeline is well explained with clear equations as well as training specifics (timings, optimizers etc.)

**Weaknesses:**

1. Limited forecasting capacity at high sparsity: The experiment results shows that at 70% sparsity the quality of LFF performance degrades notably. The author already acknowledges this but I wonder what are some potential resolution to this that you have in mind?

2. Evaluation breadth: It seems that this approach would be particularly promising for long context tasks, but the authors didn't directly benchmarked on this part. I am wondering if there is any related result on long context evaluation?

**Questions:**

1. The author mentioned that KV-cache reduction would hurt quality, I wonder if you could provide an understanding of KV memory saving vs. improvement/loss in PPL/benchmark accuracies for me to better understand this trade-off?

(See weaknesses for the rest)

---

> ### Author Response · Authors · 2025-11-24
> **Response to Reviewer gfbw**
>
> Thank you very much for your constructive comments. I will now respond to your comments and questions point by point.
>
> 1. **Issue: some potential resolution to l  imited forecasting capacity at high sparsity**
>    We believe the key point is the design of LFF. In existing experiments, we have demonstrated that under low sparsity (25%), simple LFF can fit well, so there is no need for subsequent LoRA fine-tuning to achieve considerable performance. When sparsity further increases, it gradually approaches and exceeds the fitting limit of LFF. Further experiments indicate (please refer to Official Comment: Wall-clock speed and the selection of LFF) that when we enhance the capability of LFF, we can improve the sparsity of the corresponding module tokens and the final performance. However, we must note that improving the capability of LFF almost inevitably introduces additional parameters, making the overall optimization a trade-off.
>
> 2. **Issue: Evaluation on long context tasks**
>    Experiments on long-context tasks are currently in progress, and we will update the results as soon as they are available.
>
> 3. **Issue: KV-cache reduction trade-off**
>    Below we provide specific data, including model performance and the savings in KV cache (Model：Llama3.1-8b @ 25% sparsity, batchsize:1, sequence lenth:4096).
>
> | Model       | KV Cache(GB)    | PPL@wikitext2     |
> |-------------|-----------------|-----------------------------|
> | Llama       | 2.0               |   7.33    |
> | LFF w.o KVR | 2.0               |   8.91      |
> | LFF w. KVR  |  1.5            |   11.21     |

---

> > ### Author Response · Authors · 2025-12-03
> > **Evaluation on long context tasks**
> >
> > We appreciate the reviewer's question regarding the performance of our method in long-context settings. To evaluate this, we conducted experiments on **LongBench (ACL 2024)**, focusing on three representative long-text tasks:
> > - **Long Code Completion (lcc_e)**
> > - **Passage Retrieval (passage_retrieval_e)**
> > - **Few-shot Learning (trec_e)**
> >
> > The results are summarized in the table below:
> >
> > | Task  / Metric                        | Llama3.2-3B | Greedy Routing | Informed Routing (ours) |
> > |--------------------------------|----------|-----------------------------------------|------------|
> > | Long Code Completion (lcc_e) / code_sim_score   | 0.2969  | 0.2427                                 | 0.2515    |
> > | Passage Retrieval (passage_retrieval_e) / retrieval_score | 0.1548  | 0.1128                                 | 0.1159    |
> > | Few-shot Learning (trec_e) / classification_score     | 0.6267  | 0.4800                                 | 0.5533   |
> >
> > **Key Observations and Analysis:**
> > - As noted, dynamic computation methods generally exhibit a performance drop in long-text tasks (up to 25%) compared to short-text scenarios (~10%). We attribute this gap primarily to the **training data and setup**: our router training and subsequent LoRA fine-tuning used natural language text from **RedPajama** with a maximum sequence length of **2,048 tokens**. This leads to a mismatch in both **text length** (2k vs. 4k/8k in long-context tasks) and **text type** (natural language vs. code or retrieval-focused content), which can affect generalization.
> > - Despite this, **our method consistently outperforms traditional Greedy Routing approaches** across all tasks, aligning with the conclusions from short-text experiments. The introduction of a "linear approximation" alternative mitigates information loss from skipping, providing more robust token-level decisions. We acknowledge that further optimization for long contexts—such as training on longer sequences or domain-adaptive data—could bridge the gap, and we plan to explore this in future work.

---

### Author Response · Authors · 2025-11-24
**Official Comment: Wall-clock speed and the selection of LFF**

Since multiple reviewers have emphasized the importance of the algorithm's wall-clock speed and the selection of LFF, we first address these two issues in this public comment for a unified response.

# Wall-clock Speed
We implemented two inference versions:
1. **A pure PyTorch version.** The core idea is to group tokens based on the router's decision using `torch.gather`. After separate computations through the LFF and LLM, the batch is reassembled in the original order.
2. **A customized Triton kernel** for further performance improvement (still under optimization).

## Pure PyTorch Version
We recorded the token processing speed of the proposed algorithm during the prefill and decode phases under different sparsity levels (25%, 50%, with 0% representing the original dense model) and different sequence lengths. Specifically:
- In the **prefill phase**, we report the model's token processing speed (input token length / time consumed).
- In the **decode phase**, we report the model's token generation speed.

All evaluations were conducted on the same computing platform: Intel(R) Xeon(R) w5-2445 + NVIDIA RTX 6000 Ada (48G VRAM).

| Sparsity | Seq Len | Prefill (tokens/s)     | Decode (tokens/s)      |
|----------|---------|------------------------|------------------------|
| 0%       | 1024    | 1516.36                | 1.10                   |
| 25%      | 1024    | 1704.04 (*+12.38%*)    | 1.28 (*+16.36%*)       |
| 50%      | 1024    | 2126.72 (*+40.25%*)    | 1.52 (*+38.18%*)       |
| 0%       | 2048    | 1587.74                | 0.62                   |
| 25%      | 2048    | 1742.85 (*+9.77%*)     | 0.67 (*+8.07%*)        |
| 50%      | 2048    | 2254.24 (*+41.98%*)    | 0.84 (*+35.48%*)       |
| 0%       | 4096    | 1464.77                | 0.30                   |
| 25%      | 4096    | 1686.27 (*+15.12%*)    | 0.34 (*+13.33%*)       |
| 50%      | 4096    | 2189.87 (*+49.50%*)    | 0.46 (*+53.33%*)       |

**Observations:**
- Due to repeated `gather` operations for data movement, the increased data transfer overhead reduces the acceleration effect, especially under low sparsity and short sequence lengths. For example, the actual speedup for 25% sparsity is approximately 8–16%.
- Under long sequences and high sparsity (e.g., 4096 sequence length and 50% sparsity), the algorithm achieves near-theoretical acceleration, with ~50% speedup in both prefill and decode phases.

## Customized Kernel
To reduce data movement overhead, we developed a **query-sparse attention kernel** that minimizes gather operations. We compare its throughput (TFlops) with FlashAttention under 25% sparsity:

| Seq Len | Flash-Attn (0%) | Query-Sparse-Attn (25%)     |
|---------|-----------------|-----------------------------|
| 8       | 0.002444        | 0.003242 (*+32.65%*)        |
| 16      | 0.010331        | 0.012939 (*+25.24%*)        |
| 32      | 0.040915        | 0.052178 (*+27.52%*)        |
| 64      | 0.163546        | 0.207392 (*+26.81%*)        |
| 128     | 0.652099        | 0.833692 (*+27.84%*)        |
| 256     | 2.596289        | 3.236346 (*+24.65%*)        |

**Remarks:**
- The customized kernel avoids redundant data movement and achieves theoretical acceleration even for short sequences.
- Currently, only attention sparsity is implemented; MLP sparsity kernels are under development for end-to-end evaluation.

# Selection of LFF
In our preliminary exploration, we used a simple two-layer linear network as the LFF (i.e., 4096–100–4096). We found this sufficient for moderate sparsity levels (25%–40%). However, under extreme sparsity (70%), limited fitting capacity necessitates LoRA fine-tuning for accuracy recovery, consistent with prior methods.

Given reviewer interest in LFF structure selection, we present additional exploratory results:

## Simple Non-Linear Module
Adding a non-linear layer (e.g., 4096–100–ReLU–4096) does not significantly improve token fitting. The convergence loss gap between this and the linear network is negligible.

## Slimmed LLM Module
To improve MLP token fitting, we adopted a **slimmed LlamaMLP** module with a reduced intermediate dimension (100 vs. original 4096×4). This structure:
- Doubles LFF parameters but remains lightweight.
- Enhances MLP token fitting:
  - Higher cosine similarity for MLP tokens: **0.91** (Slimmed LlamaMLP) vs. 0.89 (linear) and 0.86 (original LLM).
  - Router selects more MLP tokens for approximation: **30.5%** (Slimmed LlamaMLP) vs. 24.1% (linear).
  - Achieves lower training loss (**2.89** vs. 3.02) and perplexity (**17.99** vs. 20.49) for language modeling.

---

### Meta-Review · Area_Chair_HPz2 · 2026-01-07

**Summary:**

This paper received 4 reviews. The reviewers (score/confidence) are: 17Wo (2/5), FY6Y (6/4), gfbw (6/2), uWVk (4/2)
Their major concerns:
- Methodology:
  - Practical deployment efficiency of the informed routing paradigm in large-batch parallel inference, specifically the heterogeneous batch latency and "waiting problem" of sparse batches that may offset theoretical efficiency gains ``17Wo (2/5)``
  - Limited expressiveness of the two-layer linear LFF architecture, and the lack of exploration on non-linear LFF designs to address performance degradation at high sparsity (70%) ``FY6Y (6/4)``
  - Accumulation of approximation errors from stacked LFFs across layers and its potential negative impact on token representations in later layers ``FY6Y (6/4)``
  - Insufficient integration of LFF mechanism with KV-cache reduction, as the current masking strategy does not leverage LFF's predictive capability for approximate KV pair generation ``FY6Y (6/4)``
- Experiments:
  - Absence of quantitative analysis on wall-clock inference speedup (tokens/second or latency) and training step acceleration, which is critical to validate the claimed efficiency gains against baselines like SkipGPT ``FY6Y (6/4), uWVk (4/2)``
  - Limited evaluation scope on model scales, with no validation on larger LLMs (13B/30B/70B) where attention/FFN balance and KV-cache behavior differ significantly ``uWVk (4/2)``
  - Lack of direct benchmarking on long context tasks, despite the paradigm's potential suitability for such scenarios ``gfbw (6/2)``
  - Insufficient quantitative analysis on the trade-off between KV memory saving and PPL/accuracy changes ``gfbw (6/2)``
  - Rapid performance degradation at high sparsity ratios, which limits the framework's practical value for LLM compression ``17Wo (2/5)``

**Reviewer Concerns:**

For the two negative reviewers:

uWVk (4/2): The reviewer raised concern to suggest adding experiments of larger models. In rebuttal, the authors mentioned they lack sufficient computation resources, so this concern is not addressed. The authors' other concerns like code releasing etc are addressed.

17Wo (2/5) is the most negative reviewer. Based on the discussions, I believe his/her concerns are also outstanding.

**Reviewer Scores:**

For the two negative reviewers, uWVk may keep the score at 4 since his/her concerns are not fully addressed. 17Wo will probably keep the 2 rating since his/her initial confidence is strong and the raised concerns are indeed problems - the authors mentioned this reviewer's concern is sort of questioning the area, though.

---

### Decision · Program_Chairs · 2026-01-26

Reject